# Therapeutic Effect and Immune Changes after Treatment of *Hymenolepis nana*-Infected BALB/c Mice with Compounds Isolated from *Leucaena leucocephala*

**DOI:** 10.3390/vetsci9070368

**Published:** 2022-07-18

**Authors:** Yi-Hsuan Ma, Chung-Yi Chen, Li-Yu Chung, Chuan-Min Yen, Yung-Shun Juan, Rong-Jyh Lin

**Affiliations:** 1Department of Parasitology, School of Medicine, College of Medicine, Kaohsiung Medical University, Kaohsiung 80708, Taiwan; u102800001@kmu.edu.tw (Y.-H.M.); leyich@kmu.edu.tw (L.-Y.C.); chmiye@kmu.edu.tw (C.-M.Y.); 2Graduate Institute of Medicine, College of Medicine, Kaohsiung Medical University, Kaohsiung 80708, Taiwan; 3School of Medical and Health Sciences, Fooyin University, Daliao, Kaohsiung 83102, Taiwan; xx377@fy.edu.tw; 4Department of Urology, College of Medicine, Kaohsiung Medical University, Kaohsiung 80708, Taiwan; juanuro@gmail.com; 5Department of Urology, Kaohsiung Medical University Hospital, Kaohsiung 80756, Taiwan; 6Graduate Institute of Clinical Medicine, College of Medicine, Kaohsiung Medical University, Kaohsiung 80708, Taiwan; 7Department of Medical Research, Kaohsiung Medical University Hospital, Kaohsiung 80756, Taiwan

**Keywords:** *Hymenolepis nana*, *Leucaena leucocephala*, 13^2^-hydroxy-(13^2^-S)-pheophytin a, aristophyll-C

## Abstract

**Simple Summary:**

This article is mainly about the development of natural medicines against the zoonotic tapeworm (*Hymenolepis nana*), which is caused by tapeworm eggs in infected mice, through the isolation of the components of jumbay (*Leucaena leucocephala*). This study allowed for an understanding of the drug therapy through the testing of these isolated components in vitro and in vivo. The authors assessed the survival and motility, and they confirmed that the two components had good effects against tapeworm. One of these components has a promising therapeutic effect in tapeworm-infected mice in the response to egg production, worm numbers, and immune responses. This article highlights the immune-enhancing and therapeutic effects of 13^2^-hydroxy-(13^2^-S)-pheophytin a against tapeworms in infected mice.

**Abstract:**

Background/Purpose: *Hymenolepis nana* is globally distributed. *Leucaena leucocephala* has been studied as a treatment, including the nematodes and protozoa, but no research results are related to cestodes. Therefore, the aim of this study was to target *H. nana*. Methods: The natural components of *L. leucocephala* were isolated and added to *H. nana,* which was cultured in vitro, to observe changes in the mortality, motility, and morphology. BALB/c male mice infected with *H. nana* were treated with effective components of *L. leucocephala* for 10 days, and the changes were recorded. After the mice were sacrificed, the spleen weight was measured, and a primary culture was performed for the subsequent cytokine and chemokine testing. Results: The experiment found that 13^2^-hydroxy-(13^2^-S)-pheophytin a and aristophyll-C have clear cestocidal effects in vitro. 13^2^-hydroxy-(13^2^-S)-pheophytin a has been shown to be effective at reducing parasite populations and eliciting host immune responses in vivo. IL-2, IL-4, IL-5, IL-6, IL-10, IL-13, IL-17, MCP-1, IFN-γ, TNF-α, MIP-1α, and GM-CSF in 13^2^-hydroxy-(13^2^-S)-pheophytin a were significantly increased after stimulation, while IL-1α, IL-1β, IL-3, IL-12p70, and RANTES were unchanged. Conclusions: The investigation shows that components of *L. leucocephala* have actual cestocidal activity against *H. nana*.

## 1. Introduction

*Hymenolepis nana* is globally distributed, and it is especially common in temperate and tropical regions. According to the 2019 European Research Report census results from 1999 to 2016, parasites of the genus *Hymenolepis* are commonly present in hosts throughout Europe [1]. *H. nana* is the most common zoonotic tapeworm, and it is estimated to infect as many as 75 million people worldwide. Its prevalence in children is as high as 25% [2]. Compared with other zoonotic infections of tapeworms, *H. nana* infection does not require the assistance of an intermediate host; this feature allows *H. nana* to directly ingest the eggs for infection, and it can also cause autoinfection in the intestinal tract [3]. According to previous studies, *H. nana* can also be transmitted to rodents through fleas and beetles as intermediate hosts [4]. The transmission of *H. nana* is very common in developing and undeveloped countries, and especially in rural and suburban areas, especially among children [5]. It has been reported that 61% of patients infected with *H. nana* are also infected with other intestinal parasites [6], which makes patients more susceptible to other diseases, and makes it more difficult to recover. As a typical treatment of trematodes and cestodiasis, praziquantel is very effective in the treatment of helminths. In addition to efficacy, praziquantel’s patent has expired and the cost is very low [7]. However, praziquantel is not a perfect drug because of some potential problems, including side effects [8,9,10], which may lead to drug resistance [11], and an unsuspected response after killing the worm [12]. Based on previous studies in rats, the treatment of praziquantel was unable to completely eradicate the hymenolepiasis in rats, and follow-up fecal examinations were necessary [13].

*Leucaea* is a genus of the legume family *Fabaceae*, which is a small tree or shrub that is native to southern Mexico and northern Central America. The Dutch introduced it to Taiwan in the 1640s for its economic value. In early Taiwan, the young stems and leaves of *L. leucocephala* were used as livestock feed. It was also burned as fuel and, in the 1960s, it was made into pulp for export. In the 1980s, as Taiwanese society gradually transformed, *L. leucocephala* lost its economic value and the plant was naturalized throughout the island, where it was in serious competition with native plants.

The leaves and fruits of *L. leucocephala* contain phytoalkaloids, and the leaves of *L. leucocephala* contain toxic mimosine (Mimosine, b-[N-(3-hydroxy-4-pyridone)]-a-aminopropionicacid) and poison phenol compounds, such as acids and flavonoids [14,15].

Previous studies have pointed out that the water extracts of the leaves and seeds of *L. leucocephala* have the effect of lowering blood sugar, that the flavonoids contained in the leaves are diuretic, and that the alkaloids have the effect of protecting the liver and reducing enzymes [16]. The antioxidant activity of the green leaves, seeds, brown pods, and crude extracts of the green pods of *L. leucocephala* were evaluated, and the main components were isolated. The test results showed that the methanol crude extracts from the seeds and brown bean pods of *L. leucocephala* had better antioxidant activity. After separation and purification, two compounds, β-sitosterol (Compound **8**) and linoleic acid (Compound **1**), were identified from the seeds, and linoleic acid, lupeol (Compound **9**), and β-sitostenone (Compound **10**) were identified from the brown bean pods. In recent years, there have been more and more studies on the therapeutic potential of *L. leucocephala*. In addition to improving immunity, research on the use of *L. leucocephala* derivatives to fight microbial infections has also increased.

1, 3-dipalmitoyl-2-oleoylglycerol (Compound **11**) and 5α, 8α-epidioxy-(24ξ)-ergosta-6, 22-dien-3β-ol (Compound **7**), as well as five other compounds, were identified from immature pods, and the green leaves yielded pheophorbide a methyl ester (Compound **3**), pheophytin-a (Compound **2**), aristophyll-C (Compound **5**), 13^2^-hydroxy-(13^2^-S)-pheophytin a (Compound **4**), and methyl-13^2^-hydroxy-(13^2^-S)-pheophorbide b (Compound **6**). Table 1 shows a complete list of the 16 isolated compounds. These pure compounds have only weak antioxidant activity and tyrosinase inhibitory properties, but seven of these compounds have been shown to inhibit cancer-cell growth and exhibit pharmacological effects, according to the literature.

Recent studies have shown that it can be used to treat pathogenic fungal infections [17]. Extracts from various parts of *L. leucocephala* had therapeutic effects on parasitic nematodes found in sheep [18,19]. In addition to nematodes [20,21,22], there are also articles related to protozoa [23], but, unfortunately, no clear curative effect has been found against nematodes and cestodes. In addition to worms, components of *L. leucocephala* were also found to be effective against some insects of medical importance [24]. Based on the therapeutic effect on nematodes, *L. leucocephala* has potential in the drug development of nature products for the treatment of livestock in animal husbandry. Ingredients with special efficacy can be extracted from the leaves and fruits of *L. leucocephala* to increase its utilization. In future studies, the compounds will be further evaluated for their antiparasitic and pharmacological effects.

In this study, we selected the more effective components from the isolated components of *L. leucocephala* to evaluate the effects of *H. nana* in vitro, and we conducted more detailed concentration effects and observations. In the animal experiments, we fed active ingredients to male BALB/c mice infected with *H. nana*, and we further observed the effect of the active ingredients in *L. leucocephala* on the infection indicators.

## 2. Materials and Methods

### 2.1. Plant Material

The *L. leucocephala* sample was collected from Kenting National Park, Pingtung County, Taiwan, in May 2009. A voucher specimen was characterized by Dr. Jin-Cherng Huang of the Department of Forest Products Science and Furniture Engineering, National Chiayi University, Chiayi, Taiwan, and it was deposited in the School of Medical and Health Sciences, Fooyin University, Kaohsiung County, Taiwan.

### 2.2. Extraction and Isolation of L. leucocephala

The air-dried green beans of *L. leucocephala* (5.0 kg) were extracted with methanol (MeOH, 80 L × 6) at room temperature. The MeOH phase extract (132.5 g) was obtained upon concentration under reduced pressure, and it was chromatographed over silica gel using n-hexane/acetone as the eluent to produce 10 fractions. Part of fraction 1 (8.24 g) was subjected to Si gel chromatography by eluting with n-hexane/acetone (50:1), and it was then enriched with acetone to furnish 7 fractions (1-1~1-7). Fraction 1-1 (1.72 g) was re-subjected to Si gel chromatography and was eluted with n-hexane/acetone (80:1) to obtain linoleic acid (**1**) (43 mg). Fraction 1-4 (2.17 g) was re-subjected to Si gel chromatography and was eluted with n-hexane/acetone (40:1) to obtain β-sitostenone (**10**) (11 mg). Fraction 1-5 (1.94 g) was subjected to Si gel chromatography by eluting with n-hexane/acetone (50:1) to obtain β-sitosterol (**8**) (15 mg). Part of fraction 2 (2.67 g) was subjected to Si gel chromatography by eluting with n-hexane/acetone (50:1) to obtain ficaprenol-11 (**11**) (21 mg). Part of fraction 3 (6.77 g) was subjected to Si gel chromatography by eluting with n-hexane/acetone (8:1), and it was then enriched with acetone to furnish 5 fractions (3-1~3-5). Fraction 3-3 (1.33 g) was further purified by another silica gel column using n-hexane/acetone to obtain squalene (**12**) (17 mg) and lupeol (**9**) (24 mg). Part of fraction 5 (7.42 g) was subjected to Si gel chromatography by eluting with n-hexane/acetone (8:1) to obtain pheophorbide a methyl ester (**3**) (12 mg). Part of fraction 6 (5.31 g) was subjected to Si gel chromatography by eluting with n-hexane/acetone (8:1) to obtain methyl-13^2^-hydroxy-(13^2^-S)-pheophorbide-b (**6**) (16 mg). Part of fraction 8 (4.91 g) was subjected to Si gel chromatography by eluting with n-hexane/acetone (5:1) to obtain pheophytin-a (**2**) (22 mg).

The air-dried leaves of *L. leucocephala* (5.8 kg) were extracted with MeOH (80 L × 6) at room temperature, and the MeOH extract (143.5 g) was obtained upon concentration under reduced pressure. The MeOH extract was chromatographed over silica gel using n-hexane/acetone as an eluent to produce 8 fractions. Fraction 2 (2.97 g) was re-subjected to Si gel chromatography by eluting with n-hexane/acetone (40:1) to obtain 13^2^ -hydroxy- (13^2^ -S)-pheophytin a (**4**) (32 mg). Part of fraction 5 (9.22 g) was subjected to Si gel chromatography by eluting with n-hexane/acetone (8:1) to obtain aristophyll-C (**5**) (33 mg). Part of fraction 6 (13.89 g) was subjected to Si gel chromatography by eluting with n-hexane/acetone (4:1), and it was then enriched with acetone to obtain 5α,8α-epidioxy-(24)- ergosta-6,22-dien-3β-ol (**7**) (43 mg) and pyropheophorbide a (**15**) (17 mg). Part of fraction 8 (7.16 g) was subjected to Si gel chromatography by eluting with n-hexane/acetone (5:1) to obtain a trans-coumaric acid (**13**) and cis-coumaric acid (**14**) mixture (5 mg).

### 2.3. Experimental Procedures for L. leucocephala Components 

Optical rotations were determined with a JASCO DIP-370 digital polarimeter. UV spectra were obtained in CH_3_CN using a JASCO V-530 spectrophotometer, and IR spectra were evaluated on a Hitachi 260-30 spectrophotometer. The 1H-NMR (500 MHz, using CDCl3 as solvent for determination), 13C-NNR (125 MHz), DEPT, HETCOR, COSY, NOESY, and HMBC spectra were obtained on a Varian (Unity Plus, CA, USA) NMR spectrometer. The low-resolution electrospray ionization mass spectrometry (ESI-MS) spectra were obtained on an API 3000 (Applied Biosystems, CA, USA). Silica gel 60 (Merck, 70–230 mesh, 230–400 mesh) was used for column chromatography. Precoated silica gel plates (Merck, Darmstadt, Germany, Kieselgel 60 F254, 0.50 mm, 0.20 mm) were used for analytical thin-layer chromatography (TLC), and precoated silica gel plates (Merck, Kieselgel 60 F-254, 0.50 mm) were used for preparative TLC. Spots were detected by spraying the plates with 50% H_2_SO_4_ and then heating them on a hot plate [25,26].

### 2.4. Materials, Kits, and Chemicals

Dulbecco’s Modified Eagle Medium (12100046), RPMI-1640 (31800022), L-glutamine (A2916801), penicillin G (15140-122), streptomycin (15140-122), amphotericin B (15290-017), and all other cell-culture reagents were purchased from Gibco BRL Life Technologies (Grand Island, NY, USA). Concanavalin A (con A) (L7647) and dimethyl sulfoxide (DMSO) (D2650) were obtained from the Sigma-Aldrich Chemical Co. (St. Louis, MO, USA). The Multiplex ELISA Kit For Mouse Cytokine (MEK1006, GMCSF, IFN-γ, IL-1α, IL-1β, IL-2, IL-3, IL-4, IL-5, IL-6, IL-10, IL-12p70, IL-17, MCP-1, MIP-1α, RANTES, TNFα) was obtained from Boster Biological Technology (Pleasanton, CA, USA). Certified fetal bovine serum (FBS) (04-001-1A) was obtained from Biological Industries (Kibbutz Beit Haemek, Israel).

### 2.5. Ethics Statement

The planning and experimental operation specifications for the animals in this article were reviewed and approved by the Institutional Animal Care and Use Committee of Kaohsiung Medical University (IACUC Approval Number: KMU-M109239), in compliance with Republic of China (ROC) laws (Animal Protection Act (amended date: 29 June 2011) and Enforcement Rules of Animal Protection (announcement date: 19 January 2000)). This study received approval from the IACUC at Kaohsiung Medical University. Male BALB/c mice, weighing 25–30 g and 4–6-weeks old, were provided by the National Laboratory Animal Center (Taipei, Taiwan). The mice were raised in the Laboratory Animal Center with air-conditioned control (at 22 ± 1 °C, with a relative humidity of 50 + 10%) and illumination control (with lights on between 7:30 and 19:30). The experimental animals were raised without any restriction to food and water.

All animal work was performed in an Association for Assessment and Accreditation of Laboratory Animal Care International (AAALAC)-accredited facility.

### 2.6. Preparation of Adult H. nana Worms

For in vitro studies, wild-type mice, which were used to maintain the *H. nana* strain, were purchased from Lin’s farm in Feng-shan, Kaohsiung, Taiwan. A total of 20 confirmed infected mice were sacrificed, and the *H. nana* were removed from the gastrointestinal tract (intestine, duodenum, jejunum, and ileum). Mature *H. nana* with all four segment types and no damage were prepared for cultivation. Mature worms were treated with gentamycin at 10 mg/mL (GIBCO) for 30 min, and they were then washed in sterile filtered PBS four times. RPMI 1640 (GIBCO) with 20% FBS (Bio Industries) was prepared for the culture medium. Adult worms were moved to 24-well plates (NUNC) and incubated for 12 h at 37 °C, 5% CO_2_. Each well contained 4 intact *H. nana* worms larger than 0.5 cm in length, and the experiment was repeated twice.

### 2.7. Mortality and Motility Assessment of H. nana by Compounds Isolated from L. leucocephala 

*H. nana* were observed individually under an inverted microscope, with the subsequent discarding of those with significant damage. The mature worms were identified based on a morphological pattern with all type segments, which included the scolex, immature segments, mature segments, and gravid proglottids, and they were randomly divided into groups and placed in 24-well culture dishes (4 worms each) containing RPMI-1640 with 20% (*v/v*) FBS, at a pH of 7.4, in an atmosphere of 95% O_2_/5% CO_2_, at 37 °C. The culture media were supplemented with L-glutamine (2 mM), penicillin (100 IU/mL), streptomycin (100 mg/mL), and amphotericin B (0.25 μg/mL), and we tested concentrations of the solvent control (0.1% DMSO), linoleic acid, pheophytin-a, pheophorbide a methyl ester, 13^2^-hydroxy- (13^2^-S)-pheophytin, aristophyll-C, methyl-13^2^- hydroxy-(13^2^-S)-pheophorbide b, 5α, 8α-epidioxy-(24ξ)-ergosta-6, 22-dien-3β-ol,β-sitosterol, lupeol, β-sitostenone, ficaprenol-11, squalene, cis-coumaric acid, trans-coumaric acid, pyropheophorbide a, and 1, 3-dipalmitoyl-2-oleoylglycerol at 100 μM. The survival and mobility of the worms were observed and evaluated at different time points by stereomicroscope. In the following step, the mature worms were scored for oscillation and peristalsis by two investigators in a blinded manner for cestocidal activity. The oscillation and peristalsis evaluation methods refer to previously published literature [27].

### 2.8. Morphological Observation of the Segments of H. nana

Mature worms of *H. nana* were treated with vehicle (0.1% DMSO), 100μM 13^2^-hydroxy-(13^2^-S)-pheophytin a, and aristophyll-C for 6 h. All worms were fixed with 4% paraformaldehyde prepared in PBS solution (pH: 7.4) for 24 h, and they were observed for variation. The morphological observation of the treated worms was carried out directly under an inverted microscope.

### 2.9. In Vivo Study and Infected Evaluation in H. nana-Infected Mice

The *H. nana* strain that was used in the experiments was maintained in our laboratory by a direct cycle using male BALB/c mice. Mature worms were obtained from the intestine, duodenum, jejunum, and ileum, and eggs were collected by the artificial laceration of the gravid proglottids. Mature eggs were suspended in a NaCl solution of 0.85% and stored at room temperature. Male BALB/c mice, weighing 25–30 g and 6 weeks old, were orally inoculated with 500 eggs in 0.5 mL 0.9% NaCl via an oral stomach tube, and the experimental operation was in a biological safety cabinet of BSL-2. The infected male mice were randomly divided into three groups: the untreated infection group, the vehicle-only group, and the 13^2^-hydroxy-(13^2^-S)-pheophytin a at 0.025 g/day group. The infection and vehicle groups each contained three male mice, while the 13^2^-hydroxy-(13^2^-S)-pheophytin a group contained five male mice for follow-up studies. In order to confirm the infection by *H. nana*, the confirmation of the eggs was tested by fecal analysis postinoculation at day 10. The fecal samples and body weights of the male BALB/c mice that were confirmed to be infected with *H. nana* were collected and recorded daily to calculate the infection status and body-weight changes until sacrificed. The tests and calculations of the EPG (eggs per gram) were investigated using the Willis method [28]. The 13^2^-hydroxy-(13^2^-S)-pheophytin a was administered orally at 0.025 g/kg/day for 10 days. Ten days after the final administration, the mice were sacrificed, and the mature worms in the gastrointestinal tract were detected and counted. The cestocidal activity was shown as the number of mature worms.

### 2.10. Primary Culture of Spleen Cells, Cultivation, and Cytokine Assay

The mice of all groups were sacrificed 10 days after inoculation with *H. nana*. The spleens were removed in a sterile environment, weighed, and scratched in 2 mL DMEM. The suspensions of spleen cell were treated with 0.085 mg/mL ammonium chloride solution (NH_4_Cl) for the hemolysis of contaminated erythrocytes, and they were then washed with DMEM twice by centrifugation at 3000 rpm for 10 min at room temperature. The cell viability exceeded 95%, as determined by trypan blue exclusion. The concentrations of spleen cells were adjusted to a final concentration of 10^7^ cells/mL, with DMEM containing 10% FBS, 72μg/mL gentamycin (Sigma), and 2 × 10^−3^ M L-glutamine. The cell suspension of spleen cell was dispensed into 1 mL/well of 24-well culture plates, and it was cultivated with or without 10 μg/mL concanavalin A (con A) at 37 °C in an atmosphere containing 5% CO_2_. The supernatants without spleen cell were harvested after 24, 48, and 72 h of incubation and were stored at −20 °C until testing. The supernatants without spleen cell were assayed for GM-CSF, IFN-γ, IL-1α, IL-1β, IL-2, IL-3, IL-4, IL-5, IL-6, IL-10, IL-12p70, IL-17, MCP-1, MIP-1α, RANTES, and TNFα, tested according to the manufacturer protocols. Cytokines and chemokines were quantitated by reference to standard curves derived from known concentrations of recombinant materials, and they were indicated as the ratio versus the control group (*H. nana*-infected group without 13^2^-hydroxy-(13^2^-S)-pheophytin a, vehicle, or con A).

### 2.11. Statistical Analysis of Data

The information was shown as mean + standard deviation (SD). The statistical significances of the death rate, oscillation, and peristalsis were estimated by a one-way analysis of variance (ANOVA), followed by Dunnett’s test or the Tukey–Kramer test. LD50 was calculated by GraphPad Prism 9. Statistical differences in EPG, worm number, and cytokine excreted by con A-induced spleen cell were estimated by the Kruskal–Wallis test, followed by Dunn’s post hoc test. A *p*-value of 0.05 was considered significant. The data were analyzed by SigmaPlot Version 10.0 and SigmaStat Version 2.03 software, Chicago, IL.

## 3. Results

### 3.1. In Vitro Mobility of L. leucocephala-Purified Compound

In the first part of the experiments, the ability of the linoleic acid, pheophytin-a, pheophorbide a methyl ester, 13^2^-hydroxy-(13^2^-S)-pheophytin a, aristophyll-C, methyl-13^2^- hydroxy-(13^2^-S)-pheophorbide b, 5α, 8α-epidioxy-(24ξ)-ergosta-6, 22-dien-3β-ol,β-sitosterol, lupeol, β-sitostenone, ficaprenol-11, squalene, cis-coumaric acid, trans-coumaric acid, pyropheophorbide a, and 1, 3-dipalmitoyl-2-oleoylglycerol were tested for cestocidal effects to modify the survival of mature worms for *H. nana* (Table 1). The oscillation and peristalsis testing results were recorded at different time points (Figure 1). 

After 72 h of testing, the linoleic acid, pheophytin-a, pheophorbide a methyl ester, methyl-13^2^-hydroxy-(13^2^-S)-pheophorbide b, 5α, 8α-epidioxy-(24ξ)-ergosta-6, 22-dien-3β-ol, β-sitosterol, lupeol, β-sitostenone, ficaprenol-11, squalene, cis-coumaric acid, trans-coumaric acid, pyropheophorbide a, and 1, 3-dipalmitoyl-2-oleoylglycerol showed slight changes in activity, but not lower than 80% of the solvent group. The worms in these constituent groups displayed no deaths or damage to the body segments. Two components, 13^2^-hydroxy-(13^2^-S)-pheophytin a and aristophyll-C, had good effects in the preliminary tests, with significant decreases in activity at 24 h and 4 h, respectively, and loss of mobility and death in a short period of time.

### 3.2. 13^2^-Hydroxy-(13^2^-S)-Pheophytin a and Aristophyll-C Induced Change of Death Rate and Mobility of Oscillation and Peristalsis Activities

Next, we look in more detail at the concentrations for testing the cestocidal effect of these two ingredients. With 13^2^-hydroxy-(13^2^-S)-pheophytin a at 100 μM, the parasites began to die at 4 h, and all died by 48 h. At the concentration of 50 μM, death was delayed until 8 h, but at 24 h, all the parasites were observed to be dead. For other concentrations below 50 μM, no parasite death was observed (Figure 2A).

With aristophyll-C at the highest concentration of 100 μM, the worms began dying at 4 h, and more than 75% were dead at 8 h, nearly 90% at 12 h, and until 48 h, the death rate reached 100%. The lower concentration of 50 μM caused over half of the dead worms at 8 h, but there was no change at subsequent time points. At the concentration of 10 μM, the time point of parasite death was delayed until 12 h, but all died at 24 h. At the concentration of 5μM, the parasites also began dying at 24 h, and one half of the worms were dead at 48 h, but there was no change afterwards. The 1 μM concentration did not result in worm death at any of the time points (Figure 2B). In terms of the lethal efficacy, the −log LD50 (μM) of the 13^2^-hydroxy-(13^2^-S)-pheophytin a and aristophyll-C for 72 h were 25.36 and 4.896, respectively.

In addition, the mortality of the worms was also observed under the same conditions. The 13^2^-hydroxy-(13^2^-S)-pheophytin a showed good results against *H. nana* oscillation. At a 100 μM concentration, the motility of the worms gradually decreased at 6 h, and continuously dropped down 50% after 2 h, finally losing all their motilities at 12 h. At a 50 μM concentration, the time point of decreased activity was delayed to 12 h, and all of them lost their activity at 24 h. No more expected results were observed at other lower concentrations (Figure 3A).

The 13^2^-hydroxy-(13^2^-S)-pheophytin a also showed excellent effects on the changes in peristalsis. At a 100 μM concentration, the peristaltic response disappeared at 4 h, while it was delayed until 8 h at a 50 μM concentration. At a 5μM concentration, the worms completely lost their peristalsis at 12 h, and they expressed the same result at 10 μM and 1 μM concentrations, and, in the solvent group, at 24 h (Figure 3B).

The effect of aristophyll-C on the oscillation of *H. nana* was also demonstrated. As shown in the data, at a 100 μM concentration, the oscillation was significantly decreased in activity at 12 h, and was completely lost at 48 h. At a 50 μM concentration, there was no effect on the worms’ oscillation until 48 h, and it totally stopped at 72 h. At a 10 μM concentration, a slight alteration was recorded at 72 h (Figure 4A). No significant difference was observed in the effect of the aristophyll-C on peristalsis. The effect of each concentration was very mixed within 24 h; however, some remarkable mobility was recorded at several time points with 100 μM and 50 μM concentrations (Figure 4B).

### 3.3. Morphology 

The damages caused by 13^2^-hydroxy-(13^2^-S)-pheophytin a and aristophyll-C to different worm segments are shown in Figure 5. Under the activity of 13^2^-hydroxy-(13^2^-S)-pheophytin a, the edge segments in the scolex and neck were obviously damaged. When treated with aristophyll-C, significant irregular damage was seen, but no sapient-segment damage was observed in the scolex. The body segments affected by 13^2^-hydroxy-(13^2^-S)-pheophytin a were found to have slightly ruptured lesions at the edges, while no significant changes were observed in the aristophyll-C group (Figure 5).

### 3.4. Mouse Spleen Weight, Body Weight, and EPG

Male BALB/c mice were fed with *H. nana* eggs, and the infection was later confirmed. The mice were divided into three groups: the infected group, the vehicle group, and the group treated with 13^2^-hydroxy-(13^2^-S)-pheophytin a at 0.025 g/kg. The mice were examined for body weight and egg excretion over ten days of treatment. Then, the mice were sacrificed on the eleventh day, and the spleens were weighed and collected to extract spleen cells.

There was no significant difference in the shapes or sizes of the spleens in the 13^2^-hydroxy-(13^2^-S)-pheophytin a-treated group compared with the infected group. A slight decrease in the spleen weight was found in the 13^2^-hydroxy-(13^2^-S)-pheophytin a-treated group, but no significant difference was observed (data not presented).

In terms of the change in body weight, the mice in the 13^2^ -hydroxy-(13^2^-S)-pheophytin a group presented a slight decrease in body weight in the first two days of drug administration, but they began to recover after the sixth day. No statistical difference was shown between the infected group, vehicle group, and 13^2^-hydroxy-(13^2^-S)-pheophytin a-treated group (data not presented).

The EPG test results over the ten days of treatment showed that the infected group and vehicle group had two peaks of egg excretion on the third and sixth days, while the ovulation period in the 13^2^-hydroxy-(13^2^-S)-pheophytin a-treated group occurred a day earlier, on the second day. No obvious peaks of ovulation were recorded on the days after, until the last day (day 10) (Figure 6).

### 3.5. In Vivo Cestocidal Activity of 13^2^-Hydroxy-(13^2^-S)-Pheophytin a

The male BALB/c mice were sacrificed after ten days of treatment after being infected with *H. nana*. The small and large intestines of the mice were dissected to collect and count the worms. As shown in Figure 7, the number of worms in the 13^2^-hydroxy-(13^2^-S)-pheophytin a treatment group was significantly decreased compared with the infected group, by about 50%.

### 3.6. Cultivation of Spleen Cells for Cytokine Assay

#### 3.6.1. IL-2

The expression of IL-2 in the infected + con A group gradually decreased over time, and the magnitude of the decrease was not clear in the vehicle + con A group, while the 13^2^-hydroxy-(13^2^-S)-pheophytin a + con A-administered group had two times higher rates at 24 and 48 h, as well as an obvious reduction at 72 h, compared with the infected + con A group (Figure 8A).

#### 3.6.2. IL-4

The IL-4 expression in the infected + con A group increased over time, and it was approximately twice as high as the previous time point at 48 h, but only slightly increased at 72 h. When compared with the infected + con A group, the 13^2^-hydroxy-(13^2^-S)-pheophytin a + con A group only had half of the expression at 24 h, but it suddenly increased at 48 h, and it dropped to half at 72 h (Figure 8B).

#### 3.6.3. IL-5

There was no significant difference in the expression of IL-5 between the infected + con A group and the vehicle + con A group at any of the time points. The expression of the 13^2^-hydroxy-(13^2^-S)-pheophytin a + con A group was approximately twice as high as the infected + con A group at 24 h, it was triple at 48 h, and it continued rising at 72 h (Figure 8C).

#### 3.6.4. IL-6

The IL-6 expression showed no significant change in the infected + con A group at the three time points, while the vehicle + con A group only slightly increased at 72 h, with no significant difference from the infected + con A group. The 13^2^-hydroxy-(13^2^-S)-pheophytin a + con A group had very significant increases compared with the infected + con A group at all three time points, and they increased significantly over time (Figure 8D).

#### 3.6.5. IL-10

No IL-10 expression was detected in the infected + con A and vehicle + con A groups, while the 13^2^-hydroxy-(13^2^-S)-pheophytin a + con A group showed higher expression at the 48th and 72nd hours, and it was elevated from time to time (Figure 8E). 

#### 3.6.6. IL-13,17

The expression levels of IL-13 (Figure 8F) and IL-17 (Figure 8G) had similar results. The expression level of IL-13 did not change significantly between the infected + con A group and the vehicle + con A group at the three time points. Significant increases were observed in the 13^2^-hydroxy-(13^2^-S)-pheophytin a + con A group at 72 h.

#### 3.6.7. IFN-γ

The expression of IFN-γ was significantly increased in the 13^2^-hydroxy-(13^2^-S)-pheophytin a + con A group, especially at 48 and 72 h, and there was a threefold increase at 24 h (Figure 8H).

#### 3.6.8. MCP-1

The infected + con A group, vehicle + con A group, and 13^2^-hydroxy-(13^2^-S)-pheophytin a + con A group all showed increased MCP-1 over time. The vehicle + con A group had a smaller increase rate, while the 13^2^-hydroxy-(13^2^-S)-pheophytin a + con A group increased significantly (Figure 9A).

#### 3.6.9. TNF-α

The expression level of TNF-α showed a two- to threefold increase in the 13^2^-hydroxy-(13^2^-S)-pheophytin a + con A group compared with the infected + con A group and the vehicle + con A group (Figure 9B).

#### 3.6.10. MIP-1α and GM-CSF 

The expression levels of MIP-1α (Figure 9C) and GM-CSF (Figure 9D) were roughly similar and had a clear trend. The expression of MIP-1α in the infected + con A group showed an obvious increase over time. The vehicle + con A group was found to have lower expression levels at 24 h and more significant increases at 48 and 72 h, but it was not statistically different from the infected + con A group. Compared with the infected + con A group, the 13^2^-hydroxy-(13^2^-S)-pheophytin a + con A group had a smaller increase at 24 h, but there were significant increases at 48 and 72 h.

#### 3.6.11. Others

No significant changes were observed in the concentrations of IL-1α, IL-1β, IL-3, IL-12p70, and RANTES after detection (data not shown).

## 4. Discussion

The current investigation shows that purified compounds of *L. leucocephala* were found to have cestocidal activity against *H. nana* in vitro and in vivo.

Based on our results, we observed that the immune response of 13^2^-hydroxy-(13^2^-S)-pheophytin a on infection with *H. nana* was comprehensive and included the Th1, Th2, and Th17 pathways. The treatment of 13^2^-hydroxy-(13^2^-S)-pheophytin a had an integrative effect on the immune responses of the infected mice with *H. nana*, whereas only a few of the cytokines and chemokines tested in the experiments were unresponsive. Most of the cytokines were increased after stimulation with 13^2^-hydroxy-(13^2^-S)-pheophytin a.

Pheophytin is an important derivative of chlorophyll, and it is extremely common in many natural plant extracts. In previous studies, various isomers of pheophytin were shown to have anticancer and anti-inflammatory effects. Pheophytin-a can increase the secretion of IL-2 and IFN-γ by intestinal immune cells, which is consistent with the phenomenon observed in this study [29].

According to a previous study published in 1997, the authors found that IFN-γ, IL-2, IL-3, IL-4, and IL-5 were significantly changed in BALB/c mice after infection with a small amount of *H. nana* (100 eggs/mouse) at 14 days [30]. The data on the IFN-γ did not show after the 15th day postinfection in a previous study, but the study showed that the IFN-γ expression reappeared on the 14th day, which suggests that IFN-γ may perform in the following days. 

The literature also pointed out that IL-2 and IL-3 appeared more obvious on the 4th day after infection, and they continued that way up to 14 days [30]. Compared with our study, only a high expression level of IL-2 was observed. The expression level of IL-3 was only increased by about 50% at 20 days postinfection, and it was stimulated with con A.

Among the cytokines related to the Th2 immune pathway, our results showed a slight decrease in IL-4, while the remaining cytokines, including IL-5 and IL-13, were slightly increased.

Compared with the previous literature, IL-4 and IL-5 increased from the 4th day, reached the peak on the 5th day, and then decreased on the days after. Until the 10th day after infection, IL-4 and IL-5 were slightly recovered after hitting the low point [30]. In this study, we found that IL-4 and IL-5 still had basic expression levels after 20 days of infection. The 13^2^-hydroxy-(13^2^-S)-pheophytin a stimulated the expression of IL-5, but not IL-4.

IL-12 has not yet shown significant expression in previous studies [31], which is consistent with our observations. The explanation may be because IL-12 appears in the early stage of the immune response, and it will not be found in the later stages. The increase in the expression of IL-13 in this study is consistent with a previous study [31], and 13^2^-hydroxy-(13^2^-S)-pheophytin a could enhance the IL-13 expression. According to the other previous study, IL-4 and IL-13 can independently induce goblet-cell hyperplasia outside the Th2 immune pathway [32]. However, in this study, only the increase in the expression of IL-13 was observed, and the IL-4 expression was slightly decreased.

The expression of IL-13 was also observed in this study. IL-13 is a Th2-related cytokine that plays an important role in helminth-induced gastrointestinal immunity. The function of IL-13 is to activate goblet cells and macrophages in the gastrointestinal tract. The proliferation of goblet cells can produce many functional proteins to resist invasion by microorganisms in the gastrointestinal tract. The activation of macrophages is important for antigen presentation and pathogen clearance, both of which are important in active immunity [33].

Among the cytokines and chemokines associated with macrophages, the chemokines tested in this study included MCP-1 (CCL2), MIP-1α (CCL3), and RANTES (CCL5). The function of 13^2^-hydroxy-(13^2^-S)-pheophytin a enhanced the expressions of MCP-1 and MIP-1α, which enhanced the immunity of the macrophages in the gastrointestinal tract. Interestingly, RANTES and MIP-1α belong to similar chemokines, but only MIP-1α changes after 13^2^-hydroxy-(13^2^-S)-pheophytin a stimulation. The detailed mechanism needs more research to confirm it. Other chemokines associated with macrophages, including GM-CSF and TNF-α, were significantly increased after 13^2^-hydroxy-(13^2^-S)-pheophytin a stimulation. Taken together, 13^2^-hydroxy-(13^2^-S)-pheophytin a can enhance the activity of macrophages in BALB/c mice infected with *H. nana*.

The cytokines associated with macrophages include IL-6 and IL-10, both of which were observed to increase under the effect of 13^2^-hydroxy-(13^2^-S)-pheophytin a, which is the same trend as the chemokines. The trend of IL-1β, which is associated with macrophage phagocytosis, showed no significant changes.

The isomers of aristophyll-C and aristophyll A and B were tested in previous studies, but no response was observed for *H. nana* in vitro (data not presented). In the future, different species of parasites should be used as the research direction for aristophyll-C and its isomers.

## 5. Conclusions

This study shows the potential use of pheophytin compounds as antiparasitic drugs. Larger group studies and mechanistic studies should produce significant results for the development of natural antiparasitic drugs. We explored the effects of the isolated components of *L. leucocephala* on the survival, activity, and morphology of *H. nana*, and we further studied the therapeutic and immunological effects of 13^2^-hydroxy-(13^2^-S)-pheophytin a on *H. nana*-infected BALB/c male mice. We found that 13^2^-hydroxy-(13^2^-S)-pheophytin a has good treatment effects on *H. nana* in mice, and it can trigger the host’s immune response to fight the parasite.

## Figures and Tables

**Figure 1 vetsci-09-00368-f001:**
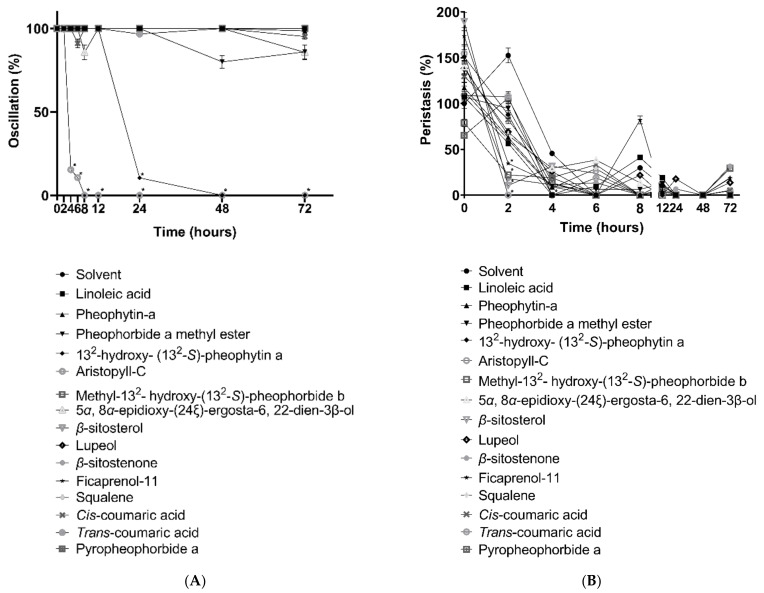
Effects of compounds isolated from *Leucaena leucocephala* on *H. nana* in the change of oscillation (**A**) and peristalsis (**B**) with linoleic acid, pheophytin-a, pheophorbide a methyl ester, 13^2^-hydroxy-(13^2^-S)-pheophytin a, aristophyll-C, methyl-13^2^-hydroxy-(13^2^-S)-pheophorbide b, 5α, 8α-epidioxy-(24ξ)-ergosta-6, 22-dien-3β-ol, β-sitosterol, lupeol, β-sitostenone, ficaprenol-11, squalene, cis-coumaric acid, trans-coumaric acid, pyropheophorbide a, and 1, 3-dipalmitoyl-2-oleoylglycerol. All components were tested in 100 μM at 2, 4, 6, 8, 12, 24, 48, and 72 h. Solvent was 0.1% DMSO. Each data point represents the mean ± SD of three individual experiments. * *p* < 0.05: significantly different from solvent.

**Figure 2 vetsci-09-00368-f002:**
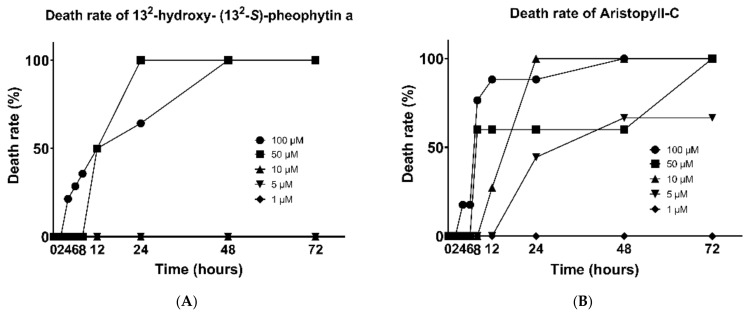
Cestocidal activity on *H. nana* of 13^2^-hydroxy-(13^2^-S)-pheophytin a (**A**) and aristophyll-C (**B**). Treatment with 1, 5, 10, 50, and 100 μM, with incubation times of 2, 4, 6, 8, 12, 24, 48, and 72 h.

**Figure 3 vetsci-09-00368-f003:**
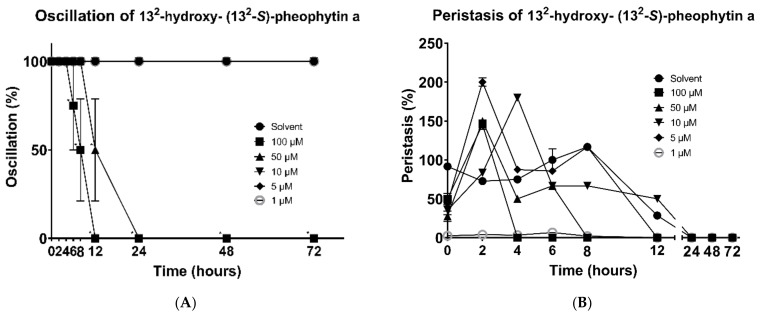
Effects of 13^2^-hydroxy-(13^2^-S)-pheophytin a on *H. nana*. Treatment with various concentrations of 13^2^-hydroxy-(13^2^-S)-pheophytin a (1, 5, 10, 50, and 100 μM), with incubation times of 2, 4, 6, 8, 12, 24, 48, and 72 h. Time course of 13^2^-hydroxy-(13^2^-S)-pheophytin a effect on *H. nana* oscillation (**A**) and peristalsis (**B**) are presented as percentages. Solvent was 0.1% DMSO. Each value is presented as the mean ± SD of three individual experiments; * *p* < 0.05 indicates a significant difference versus vehicle-treated worms.

**Figure 4 vetsci-09-00368-f004:**
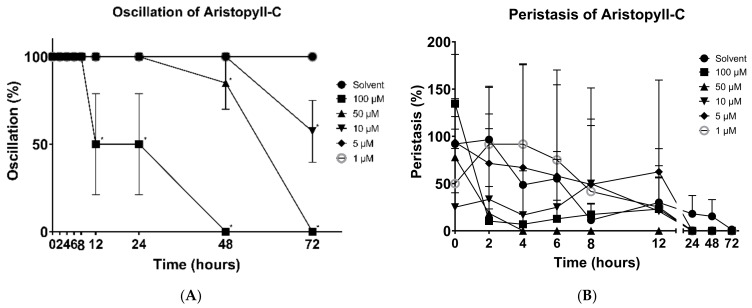
Effects of aristophyll-C on *H. nana*, treated with different concentrations of aristophyll-C (1, 5, 10, 50, and 100 μM), with incubation times of 2, 4, 6, 8, 12, 24, 48, and 72 h. Time courses of effects on oscillation (**A**) and peristalsis (**B**) are presented as percentages. Solvent was 0.1% DMSO. Each value is the mean ± SD of three individual experiments; * *p* < 0.05 indicates a significant difference from the result for vehicle-treated worms.

**Figure 5 vetsci-09-00368-f005:**
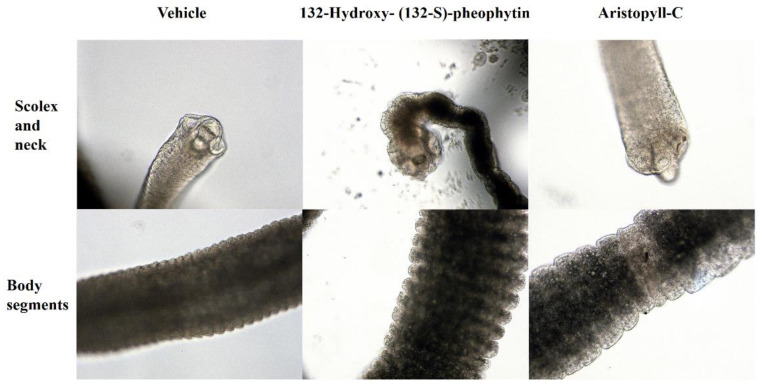
Morphology of *H*. *nana* segments at 6 h of drug treatment. *H. nana* was incubated with 0.1% DMSO as vehicle, 13^2^-hydroxy-(13^2^-S)-pheophytin a, and aristophyll-C at 100 μM, and the images were collected at 200× magnification.

**Figure 6 vetsci-09-00368-f006:**
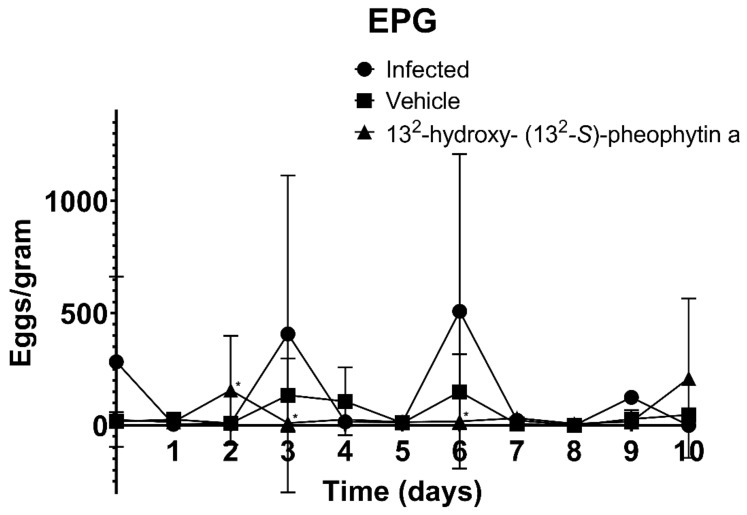
EPG examination of male BALB/c mice infected by *H*. *nana* for 10 days over treatment period. After infection, 13^2^-hydroxy-(13^2^-S)-pheophytin a was administered orally at 0.025 g/kg/day for 10 days, and a comparison was made with the infected group. * *p* < 0.05 indicates a significant difference versus the infected group.

**Figure 7 vetsci-09-00368-f007:**
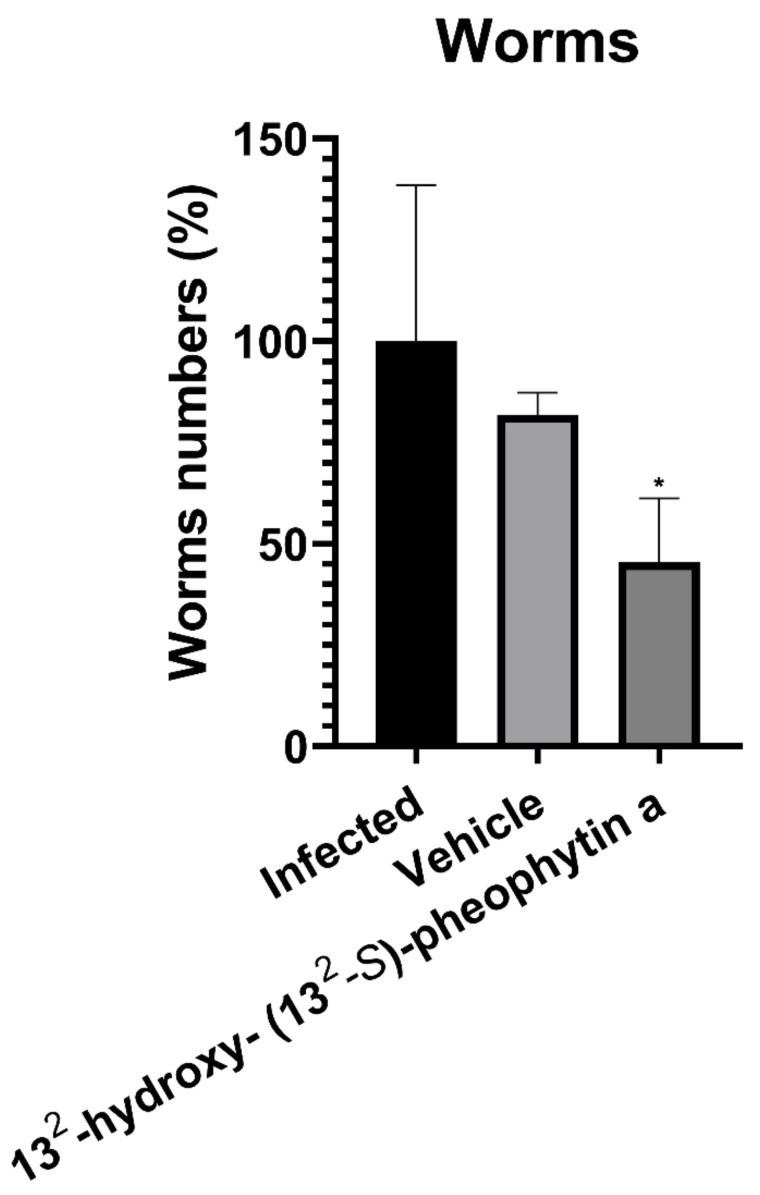
Number of *Hymenolepis nana* worms after 10 days of treatment. Male BALB/c mice were fed with 500 eggs, and we observed the feces after 10 days of infection. After the eggs appeared in the feces, 13^2^-hydroxy-(13^2^-S)-pheophytin a was administered orally at 0.025 g/kg/day for 10 days, and a comparison was made with the infected group. * *p* < 0.05 indicates a significant difference versus the infected group.

**Figure 8 vetsci-09-00368-f008:**
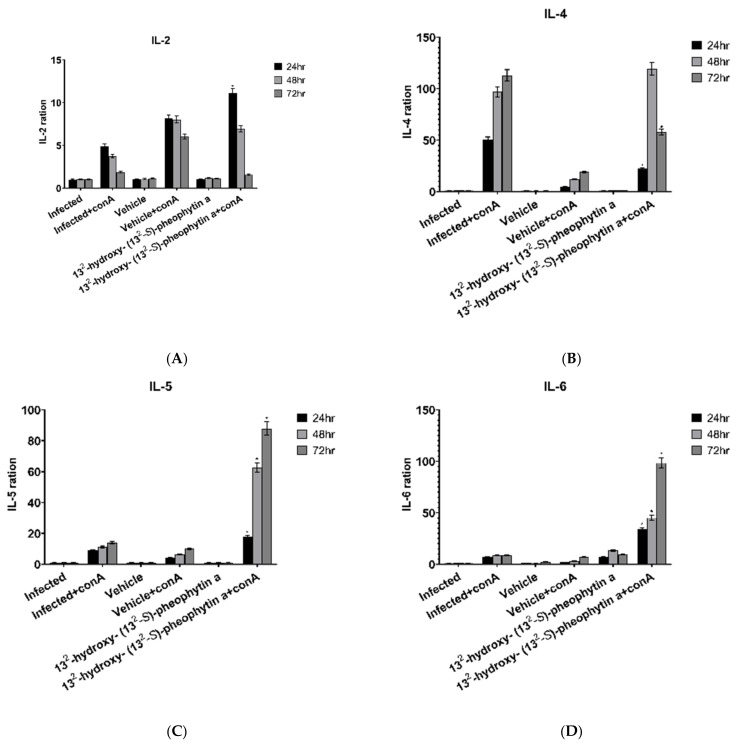
Cytokine production in spleen cells postinfected with *H. nana* for 10 days. Cell media of spleen cells were harvested to determine IL-2, IL-4, IL-5, and IL-6 after 24, 48, and 72 h incubation. Infected group: male BALB/c mice infected by *H. nana* without treatment in vivo. Vehicle and 13^2^ -hydroxy- (13^2^ -S)-pheophytin a group: infected groups were treated with vehicle or 13^2^ -hydroxy- (13^2^ -S)-pheophytin a at 0.025 g/kg/day for 10 days in vivo. Con A: spleen cells were cultured with concanavalin A. IL-2, IL-4, IL-5, and IL-6 ratios are shown at 24, 48, and 72 h of incubation (**A**–**D**). * *p* < 0.05 indicates a statistical significance from infected + con A. Cytokine production in spleen cells postinfected with *H. nana* for 10 days. Cell media of spleen cells were harvested to determine IL-10, IL-13, IL-17, and IFN-γ after 24, 48, and 72 h incubation. Infected group: male BALB/c mice infected by *H. nana* without treatment in vivo. Vehicle and 13^2^-hydroxy-(13^2^-S)-pheophytin a group: infected groups were treated with vehicle or 13^2^-hydroxy-(13^2^-S)-pheophytin a at 0.025 g/kg/day for 10 days in vivo. Con A: spleen cells were cultured with concanavalin A. IL-10, IL-13, IL-17, and IFN-γ ratios are shown at 24, 48, and 72 h of incubation (**A**–**H**). * *p* < 0.05 indicates a statistical significance from infected + con A.

**Figure 9 vetsci-09-00368-f009:**
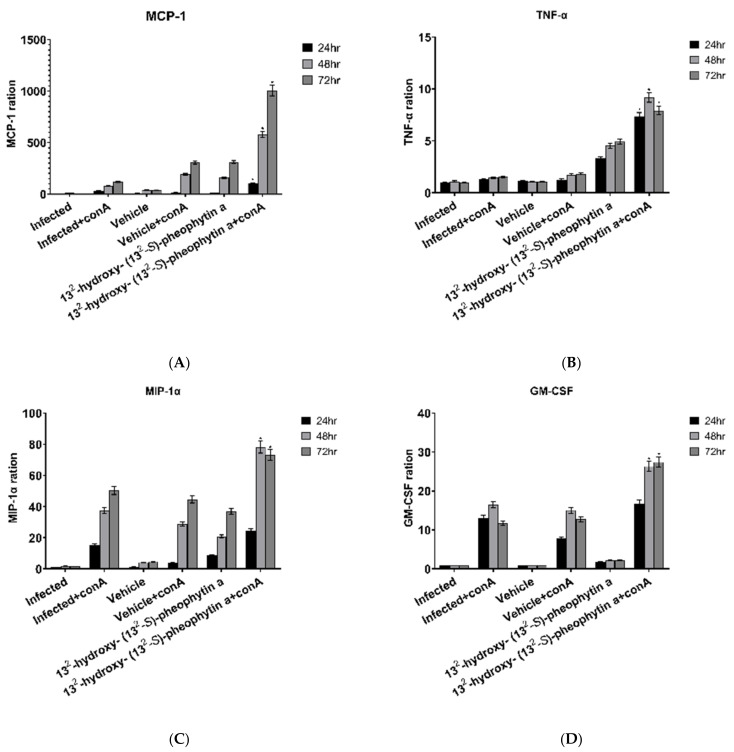
Chemokine production in spleen cells postinfection with *H. nana* for 10 days. Cell media of spleen cells were harvested to determine MCP-1, TNF-α, MIP-1α, and GM-CSF after 24, 48, and 72 h incubation. Infected group: male BALB/c mice infected by *H. nana* without treatment in vivo. Vehicle and 13^2^-hydroxy-(13^2^-S)-pheophytin a group: infected groups were treated with vehicle or 13^2^-hydroxy-(13^2^-S)-pheophytin a at 0.025 g/kg/day for 10 days in vivo. Con A: spleen cells were cultured with concanavalin A. MCP-1, TNF-α, MIP-1α, and GM-CSF ratios are shown at 24, 48, and 72 h of incubation (**A**–**D**). * *p* < 0.05 indicates a statistical significance from infected + con A group.

**Table 1 vetsci-09-00368-t001:** A total of 16 compounds were isolated from different parts of *L. leucocephala*.

Compounds	Parts
Linoleic acid (**1**)	Seeds, mature pods, immature pods
Pheophytin-a (**2**)	Leaves
Pheophorbide a methyl ester (**3**)	Leaves, immature pods
13^2^-hydroxy- (13^2^-*S*)-pheophytin a (**4**)	Leaves
Aristophyll-C (**5**)	Leaves
Methyl-13^2^- hydroxy-(13^2^-*S*)-pheophorbide b (**6**)	Leaves
5*α*, 8*α*-epidioxy-(24ξ)-ergosta-6, 22-dien-3β-ol (**7**)	Mature pods
*β*-sitosterol (**8**)	Seeds
Lupeol (**9**)	Mature pods
*β*-sitostenone (**10**)	Mature pods
Ficaprenol-11 (**11**)	Immature pods
Squalene (**12**)	Immature pods
*Cis*-coumaric acid (**13**)	Immature pods
*Trans*-coumaric acid (**14**)	Immature pods
Pyropheophorbide a (**15**)	Immature pods
1, 3-Dipalmitoyl-2-oleoylglycerol (**16**)	Mature pods, immature pods

## Data Availability

The data that support the findings of this study are available from the corresponding author upon reasonable request.

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
