# Peer review of "Therapeutic Effect and Immune Changes after Treatment of Hymenolepis nana-Infected BALB/c Mice with Compounds Isolated from Leucaena leucocephala"

_vetsci, 2022, doi:10.3390/vetsci9070368_

Round 1

Reviewer 1 Report

The article presented for review is quite interesting. The authors made an attempt to determine the therapeutic effect of some Leucaena leucocephala compounds on Hymenolepis nana-infected mice. The article is fairly well written. It can be accepted for publication after minor corrections.

Some notes:

L21, 24, 29, 35: remove space before colon.

L89: medical arhropods? What the authors meant?

L90: “very high potential” – avoid valuation (the whole manuscript). High potential is only probable.

L91: “ecomonically valuable animals” - every animal has some economic value.

L108: MeOH – explain abbreviation.

L165: Clearly indicate the number of mice used in the experiments. Clearly indicate the number of H. nana used in the experiments.

L181: correct the degree sign (the whole manuscript).

L215: discovered? What the authors meant? Detected? Found?

L235: Indicate how the normality of the distribution was checked. Indicate how the equality of variance was checked.

L255 (and the whole manuscript): graphs are unreadable (to small, to many data). Clearly indicate which data were compared.

Author Response

Response to Reviewer 1 Comments

The article presented for review is quite interesting. The authors made an attempt to determine the therapeutic effect of some Leucaena leucocephala compounds on Hymenolepis nana-infected mice. The article is fairly well written. It can be accepted for publication after minor corrections.

Figure 4(B) has been re-statistical and plotted because of errors in previous data entry and handover.

Point 1: L21, 24, 29, 35: remove space before colon.

Response 1: According to your reminder, the formatting of punctuation has been corrected.

Point 2: L89: medical arhropods? What the authors meant?

Response 2: Thanks for the reminder, the term is correctly described as "insects of medical importance". (L110)

Point 3: L90: “very high potential” – avoid valuation (the whole manuscript). High potential is only probable.

Response 3: We agree your suggestion. This paragraph of text has been revised to a more appropriate description.

Point 4: L91: “ecomonically valuable animals” - every animal has some economic value.

Response 4: Thanks for the reminder, the term is correctly described as "livestock in animal husbandry". (L112)

Point 5: L108: MeOH – explain abbreviation.

Response 5: Thank you for your suggestion. A description of methanol has been added. (L131)

Point 6: L165: Clearly indicate the number of mice used in the experiments. Clearly indicate the number of H. nana used in the experiments.

Response 6: Thank you for your suggestion. The number of H. nana worms used in the experiment and the number of mice in each group have been added in the article. (L198, 205, 213 and 241)

Point 7: L181: correct the degree sign (the whole manuscript).

Response 7: According to your reminder, the formatting of punctuation has been corrected. (L203, 210, 218 and 246)

Point 8: L215: discovered? What the authors meant? Detected? Found?

Response 8: Thank you for your suggestion. This paragraph of text has been revised to a more appropriate description.

Point 9: L235: Indicate how the normality of the distribution was checked. Indicate how the equality of variance was checked.

Response 9: Thank you for your suggestion. The description of Dunn post hoc test has been added in the paragraph. (L280)

Point 10: L255 (and the whole manuscript): graphs are unreadable (to small, to many data). Clearly indicate which data were compared.

Response 10: Thank you for your suggestion. We have tried our best to modify the image content and description into a more readable form and description.

Reviewer 2 Report

The authors describe effects of components of Leucaea leicocephala on Hymenolepis nana tapeworms in vitro and in vivo. Effects are related to mortality, motility and morphology. There are, however, major deficiencies in the presentation of the materials and methods, the results and the discussion which argue against a publication in Veterinary Sciences.

The method used for extraction of the pharmacologically active substances from Leucaea leucocephala is unclear and not reproducible. More important, however, is that the authors do not give any evidence for the identity of the extracts obtained from Leucaea leucocephala. The question arises how the authors know that they are in fact handling with the substances in purity named here. It remains an obstacle how the authors calculated the molarity of their solutions. Well characterized components, however, are the basic and of major relevance for a study and the conclusions as they are presented here.

In vitro investigations:
The exact number of worms per group is not given. The authors do not describe how mortality/survival was examined and how the time of death was determined. The authors describe in the Materials and methods section (line 190) that the investigation of oscillation and peristalsis was performed in a blinded matter by two investigators, but they do not present any results referring to this. They do not explain how oscillation and peristalsis values and percentages were determined but relate to a reference. In the result section a wobble and creep test is mentioned for the first time, but it is not explained. In the legend of figures 1, 3 and 4 it is stated that each value is presented as the mean ± SD of three individual experiments. Since a SD bar is only visible for a few values in Figures 1 and 3 and not for all values in Figure 4, this statement is questionable. As mentioned already, the number of worms included in each experimental group is also not given but would be important.

 In vivo investigations:
Three male infected BALB/c mice, each, were used in the infected/non-treated group and the vesicle-only group, respectively, while 5 mice were included in the group treated with 132-hydroxy- (132-S)-pheophytin at a dose of 0.025 g/kg and day. Infection status of the mice was evaluated at day 10 after infection, and at that day, treatment for 10 days started. During these 10 days of treatment, EPG was determined. The EPG values were fluctuating in all the groups, but the authors do not discuss this with regard to a treatment effect which is not obvious to my opinion. The mice were sacrificed at the last day of the treatment period. The percentage of mature worms was lower in the treatment group than in the not-treated group, but, here, as well, the absolute number of worms is not given. It would additionally be important to know the number of mice per group included in the evaluation and the number of repetitions of the experiments to have an information about the significance of the results.
Hymenolepis nana is the only nematode able to perform autoinfection and infect the same host without using an intermediate host. Autoinfection occurs in many mouse strains. BALB/c mice rapidly develop immunity against infection with Hymenolepis nana, and autoinfection does not seem to play a major role in this mouse strain. Both phenomena, autoinfection and immunity, are not discussed with regard to the treatment strategy proposed here.

Cytokine and chemokine assays:
The authors describe in a very detailed way the results they obtained, but a discussion with other papers of this topic (for example Conchedda et al Int J Parasitol 1998) is not performed.

Author Response

Response to Reviewer 2 Comments

The authors describe effects of components of Leucaea leicocephala on Hymenolepis nana tapeworms in vitro and in vivo. Effects are related to mortality, motility and morphology. There are, however, major deficiencies in the presentation of the materials and methods, the results and the discussion which argue against a publication in Veterinary Sciences.

Figure 4(B) has been re-statistical and plotted because of errors in previous data entry and handover.

Point 1: The method used for extraction of the pharmacologically active substances from Leucaea leucocephala is unclear and not reproducible. More important, however, is that the authors do not give any evidence for the identity of the extracts obtained from Leucaea leucocephala. The question arises how the authors know that they are in fact handling with the substances in purity named here. It remains an obstacle how the authors calculated the molarity of their solutions. Well characterized components, however, are the basic and of major relevance for a study and the conclusions as they are presented here.

Response 1: Thank you for your suggestion. Based on your suggestions, we have added references in the Materials and Methods of 2.03 Experimental procedures for L. leucocephala components (L159-171). Methods of extraction, fractionation and chemical structure identification according previously paper for co-authors Chen CY as below:

  1. Chen CY*, Wang YD. Secondary metabolites from Leucaena leucocephala. Chem. Nat. Compd., 2011, 47, 145-146. (SCI)
  1. She LC, Liu CM, Chen CT, Li HT, Li WJ,Chen CY*. The anti-cancer and anti-metastasis effects of phytochemical constituents from Leucaena leucocephala Res., 2017, 28(7), 2893-2897. (SCI)

Point 2: The exact number of worms per group is not given. The authors do not describe how mortality/survival was examined and how the time of death was determined. The authors describe in the Materials and methods section (line 190) that the investigation of oscillation and peristalsis was performed in a blinded matter by two investigators, but they do not present any results referring to this. They do not explain how oscillation and peristalsis values and percentages were determined but relate to a reference. In the result section a wobble and creep test is mentioned for the first time, but it is not explained. In the legend of figures 1, 3 and 4 it is stated that each value is presented as the mean ± SD of three individual experiments. Since a SD bar is only visible for a few values in Figures 1 and 3 and not for all values in Figure 4, this statement is questionable. As mentioned already, the number of worms included in each experimental group is also not given but would be important.

Response 2:

  1. We agree your suggestion. The number of H. nana worms used in the experiment have been added in the article. (L210, 218)
  2. Mortality monitoring of parasites was performed after imaging recordings of oscillation and peristalsis at each time point. Under the observation of an inverted microscope, gently tap the 24 well plate with finger and confirm whether the worm has a change in motility when subjected to external stimuli. We will repeat this behavior three times within one minute, and the unresponsive worm during this period will be regarded as dead. No errors of judgment occurred in the current experiment.
  3. The detailed steps and method references related to oscillation and peristalsis have been added to the article.

  1. Rong-Jyh Lin, Chung-Yi Chen, Chin-Mei Lu, Yi-Hsuan Ma, Li-Yu Chung, Jiun-Jye Wang, June-Der Lee, Chuan-Min Yen. Anthelmintic constituents from ginger (Zingiber officinale) against Hymenolepis nana, Acta Tropica, 2014 Dec;140:50-60. doi: 10.1016/j.actatropica.2014.07.009.

  1. Thank you for your suggestion. Most of the values in the graph have SD representation, but unfortunately the representation of the values is too low to be displayed in the graph, we have tried to adjust the presentation of the graph to make it easier to read.

Point 3: In vivo investigations:
Three male infected BALB/c mice, each, were used in the infected/non-treated group and the vesicle-only group, respectively, while 5 mice were included in the group treated with 132-hydroxy- (132-S)-pheophytin at a dose of 0.025 g/kg and day. Infection status of the mice was evaluated at day 10 after infection, and at that day, treatment for 10 days started. During these 10 days of treatment, EPG was determined. The EPG values were fluctuating in all the groups, but the authors do not discuss this with regard to a treatment effect which is not obvious to my opinion. The mice were sacrificed at the last day of the treatment period. The percentage of mature worms was lower in the treatment group than in the not-treated group, but, here, as well, the absolute number of worms is not given. It would additionally be important to know the number of mice per group included in the evaluation and the number of repetitions of the experiments to have an information about the significance of the results.
Hymenolepis nana is the only nematode able to perform autoinfection and infect the same host without using an intermediate host. Autoinfection occurs in many mouse strains. BALB/c mice rapidly develop immunity against infection with Hymenolepis nana, and autoinfection does not seem to play a major role in this mouse strain. Both phenomena, autoinfection and immunity, are not discussed with regard to the treatment strategy proposed here.

Response 3:

  1. Thank you for your suggestion. All ingredients in this study were extracted from naturally obtained Leucaea leucocephala. The yields of the two components found to be effective in this article, 132 -hydroxy- (132 -S)-pheophytin a and aristopyll-C, obtained after extraction, are very rare. After a drug dose evaluation, it was found that 132 -hydroxy- (132 -S)-pheophytin a could be tested on about five mice, while aristopyll-C could only be given to one mouse for study. We reduced the number of experimental animals in the Infected and Vehicle groups according to IACUC recommendations.
  2. The first assessments and ideas about EPG originated in a 2009 article on Hymenolepis diminuta. In this article, rats were used as an experimental model, and EPG was tested on days 0, 4, 14, and 24 to observe the changes of eggs in the control group and the treatment group. In this study, praziquantel, the designated drug for tapeworms, was used for treatment, but egg production has not been completely eliminated. So we thought about whether we could learn more about egg changes during treatment through more intensive observations, and we designed our experiments with that idea in mind. In a previous study we found that BALB/c mice infected with H. nana would have two peaks of ovulation within ten days, but we were unable to find enough literature to compare either the interval between the experimental tests or the strains of mouse.

  1. Meltem Uluta Esatgil, Aynur Gülanber and and Handan Aydın. Efficacy of praziquantel (injection formula) in the treatment of Hymenolepis diminuta infection in laboratory rats by oral application. Tropical Medicine and Health Vol. 37 No. 1, 2009, pp. 13-16.

  1. In the design of data presentation, we believe that the interpretation of percentages is more suitable.
  2. Thank you for your suggestion. In previous literature studies, we did know that H. nana may autoinfection in other strains of mice. In our study, we prefer to explore tests that investigate infection rates and immune responses in the absence of autoinfection. According to the previous discussion on autoinfection, it has been mentioned that in addition to the strain, the host's immunity and environment are also influencing factors. The cages and bedding of the mice were changed daily to avoid the effects of autoinfection.

Point 4: Cytokine and chemokine assays:
The authors describe in a very detailed way the results they obtained, but a discussion with other papers of this topic (for example Conchedda et al Int J Parasitol 1998) is not performed.

Response 4: We agree your suggestion. A discussion of cytokines and chemokines has been added to this article. (L530-577)

Reviewer 3 Report

In the current study the authors have tried to study the therapeutic effect immune changes of compounds isolated from Leucaena leucocephala on Hymenolepis nana.  Overall, the study is properly designed, and data seem to support authors claims. However, the authors need to perform significant revision and address the following points before its publication in Veterinary Sciences.

1.     Some of the sentences should be rephrased to improve clarity. For example, it is difficult understand the point authors are trying make with the first sentence “The prevalence of Hymenolepis nana is a global distribution.” (Line 21)

2.     The authors need to include more background on the health effect of H.nana and why this study is important.

3.  Currently available treatments for H.nana infestation should also be discussed in the background section.

4.     The authors need to explain the difference between Oscillation and peristalsis 250 and the wobble and creep test and what these different test results imply.

5.     The authors have quoted that “Oscillation and peristalsis 250 and wobble and creep test results were recorded at different time points (Figure 1).” However, only the oscillation and Perstalisis are shown in Figure 1.

6.     3.01 heading mentions cestocidal activity but showed mobility results and it is the other way in 3.02. The authors need to make sure the heading matches the content.

7.     In terms of the Death rate the standard practice is to show the LD50 values. This gives values that could be compared with other compounds in this article and published literature. Hence the authors need to show the LD50 for all the compounds rather than just the percentage.

8.     The authors need to show whether the inference from morphology was statistically different, In other words, how many H.nana was used to image in each condition to reach this inference?

Author Response

Response to Reviewer 3 Comments

In the current study the authors have tried to study the therapeutic effect immune changes of compounds isolated from Leucaena leucocephala on Hymenolepis nana.  Overall, the study is properly designed, and data seem to support authors claims. However, the authors need to perform significant revision and address the following points before its publication in Veterinary Sciences.

Figure 4(B) has been re-statistical and plotted because of errors in previous data entry and handover.

Point 1: Some of the sentences should be rephrased to improve clarity. For example, it is difficult understand the point authors are trying make with the first sentence “The prevalence of Hymenolepis nana is a global distribution.” (Line 21)

Response 1: We agree your suggestion. The manuscript has been completely re-examined and revised, and the marked sentences have been revised to make it easier to read. (Line 50)

Point 2: he authors need to include more background on the health effect of H. nana and why this study is important.

Response 2: We agree your suggestion. Narratives and literature citations regarding co-infection of H. nana with other parasites have been added to the article. (Line 61-63)

Point 3: Currently available treatments for H. nana infestation should also be discussed in the background section.

Response 3: We agree your suggestion. More descriptions of the advantages and disadvantages of praziquantel have been added to the manuscript. (Line 63-69)

Point 4: The authors need to explain the difference between Oscillation and peristalsis 250 and the wobble and creep test and what these different test results imply.

Response 4: We agree your suggestion. In the in vitro culture of H. nana, we observed through the filmed video, we observed that the segments of tapeworms have two main movement modes, oscillation and peristalsis in the environment of in vitro culture. The parasitic site of H. nana was observed in the posterior segment of the host's small intestine, and it was difficult to observe the motility and survival status without labeling in vivo. Oscillation and peristalsis are the most easily observed in the movement patterns of H. nana. Oscillation is presumed to be the behavior of the worm actively sensing the intestinal wall in the small intestine, while peristalsis is presumed to be the normal response of the worm's motility. Under the condition of in vitro culture, H. nana has been observed to have a large amount of body twisting. When the activity is extremely strong, this will greatly interfere with the observation of oscillation and peristalsis.

Point 5: The authors have quoted that “Oscillation and peristalsis 250 and wobble and creep test results were recorded at different time points (Figure 1).” However, only the oscillation and Perstalisis are shown in Figure 1.

Response 5: According your mention, the incorrectly described sentence has been corrected.

Point 6: 3.01 heading mentions cestocidal activity but showed mobility results and it is the other way in 3.02. The authors need to make sure the heading matches the content.

Response 6: We agree your suggestion. The heading of 3.01 and 3.02 has been revised to be closer to the description of the figure. (L283 and 310)

Point 7: In terms of the Death rate the standard practice is to show the LD50 values. This gives values that could be compared with other compounds in this article and published literature. Hence the authors need to show the LD50 for all the compounds rather than just the percentage.

Response 7: We agree your suggestion. LD50 calculations and results have been added to the article. (Line 277-278 and 325-327)

Point 8: The authors need to show whether the inference from morphology was statistically different, In other words, how many H. nana was used to image in each condition to reach this inference?

Response 8: Thanks for your suggestion. In the morphological experiments, 5 worms were cultured and treated in different wells of 12 well plates. Damage to worms was observed on all worms. All injuries were confirmed not to be caused by surgical instruments during the experimental transfer.

Round 2

Reviewer 3 Report

The first comment on “Some of the sentences should be rephrased to improve clarity. For example, it is difficult to understand the point authors are trying to make with the first sentence “The prevalence of Hymenolepis nana is a global distribution.” (Line 21) is still not addressed.

LD50 value was requested, however, the methodology, data, and graph are missing in the final article.

Also, the Line numbers which the authors provide in the comments almost always do not match, making it further difficult to evaluate.

Statistical analysis on the morphological evaluation was not carried out, which is one of the comments in my first report.